# Clinical efficacy and biomarker analysis of dual PD-1/CTLA-4 blockade in recurrent/metastatic EBV-associated nasopharyngeal carcinoma

Darren Wan-Teck Lim [1,2,3] ✉, Hsiang-Fong Kao[4,5], Lisda Suteja [1,2], Constance H. Li[1,2], Hong Sheng Quah [1,2], Daniel Shao-Weng Tan [1,2,6], Sze-Huey Tan [1,2], Eng-Huat Tan[1], Wan-Ling Tan[1], Justina Nadia Lee[3], Felicia Yu-Ting Wee[3], Amit Jain[1], Boon-Cher Goh[7], Melvin L. K. Chua [1,2], Bin-Chi Liao[4,5], Quan Sing Ng[1], Ruey-Long Hong[4,5], Mei-Kim Ang[1], Joe Poh-Sheng Yeong [3,8] & N. Gopalakrishna Iyer [1,2,8] ✉

Single-agent checkpoint inhibitor (CPI) activity in Epstein-Barr Virus (EBV) related nasopharyngeal carcinoma (NPC) is limited. Dual CPI shows increased activity in solid cancers. In this single-arm phase II trial (NCT03097939), 40 patients with recurrent/metastatic EBV-positive NPC who failed prior chemotherapy receive nivolumab 3 mg/kg every 2 weeks and ipilimumab 1 mg/kg every 6 weeks. Primary outcome of best overall response rate (BOR) and secondary outcomes (progression-free survival [PFS], clinical benefit rate, adverse events, duration of response, time to progression, overall survival [OS]) are reported. The BOR is 38% with median PFS and OS of 5.3 and 19.5 months, respectively. This regimen is well-tolerated and treatment-related adverse events requiring discontinuation are low. Biomarker analysis shows no correlation of outcomes to PD-L1 expression or tumor mutation burden. While the BOR does not meet pre-planned estimates, patients with low plasma EBV-DNA titre (<7800 IU/ml) trend to better response and PFS. Deep immunophenotyping of pre- and on-treatment tumor biopsies demonstrate early activation of the adaptive immune response, with T-cell cytotoxicity seen in responders prior to any clinically evident response. Immune-subpopulation profiling also identifies specific PD-1 and CTLA-4 expressing CD8 subpopulations that predict for response to combined immune checkpoint blockade in NPC.

Immune checkpoint blockade (ICB) represents a paradigm shift in the management of solid tumors, but its promise and potential remain partially realized due to a paucity of biomarkers for accurate case selection. Virally driven cancers introduce an additional level of complexity as the virus itself can be a putative target and biomarker, in addition to tumor cells and the associated immune micro-environment. Nasopharyngeal carcinoma (NPC) is endemic in South-East Asia and Southern China, and associated with

[1]National Cancer Centre Singapore, Singapore, Singapore. [2]Duke-NUS Medical School, Singapore, Singapore. [3]Institute of Molecular and Cell Biology, A*STAR, Singapore, Singapore. [4]National Taiwan University Hospital, Taipei, Taiwan. [5]National Taiwan University Cancer Center, Taipei, Taiwan. [6]Genome Institute of Singapore, A*STAR, Singapore, Singapore. [7]National University Health System, Singapore, Singapore. [8]Singapore General Hospital, Singapore, Singapore. ✉e-mail: darren.lim.w.t@singhealth.com.sg; gmsngi@nus.edu.sg

Fig. 1 | Details of clinical trial and patient cohort. a Overview of the trial schema. b Consort diagram of trial.

Epstein-Barr virus (EBV) infection[1,2]. It is highly responsive to radio-therapy and chemotherapy[3], and concurrent chemotherapy and radiotherapy is the standard of care for locally advanced disease[4–6]. Up till 2022, gemcitabine and cisplatin (GC) was standard first line treatment for recurrent/metastatic (R/M) NPC[7]. However, this is likely to change pending regulatory approvals of PD-1 inhibitors from China. Pivotal phase III studies utilizing a GC backbone and a partner PD-1 inhibitor have demonstrated superior progression-free survival (PFS) benefit over GC and placebo, although overall survival (OS) remains immature[8,9].

Recommendations for subsequent salvage chemotherapy at progression have remained unchanged for the last three decades[10] despite short median PFS and OS[11]. The introduction of single agent PD-1/PD-L1 inhibitors in the salvage setting achieved overall response rates over 20%[12–15]. However, unlike in R/M head and neck squamous cancers[16,17], this modest efficacy in NPC has not resulted in a survival benefit compared to chemotherapy[18,19]. However, it was observed that while single agent spartalizumab (PD-1 inhibitor) was not superior to combination chemotherapy[20], nonetheless up to 17% of patients ran-domized to spartalizumab achieved durable control of disease lasting beyond 12 months. The ability to pre-determine this cohort upfront using a biomarker of response would be critical in identifying patients who would benefit from ICB-therapy[19]. PD-L1 expression does not correlate well with response to PD-1/PD-L1 blockade in NPC even though it is ubiquitously expressed[15,21], unlike in non-small cell lung or head and neck squamous cell cancers[22,23]. Moreover, genomic char-acterization of NPC reveals a relatively bland mutational landscape

with low tumor mutation burden (TMB) and no dominant oncogenic drivers[24], excluding these features as biomarkers of response.

Recent data from other solid tumors suggest that dual PD-1/CTLA-4 inhibition may improve on single agent activity by promoting the mobilization of peripheral T-cells and downregulation of resident regulatory T-cells (Tregs)[25]. The latter is especially relevant in NPC, which is known to be richly infiltrated by lymphocytes comprising CD8, CD4 and a significant FOXP3-positive Treg population[26]. Com-bination PD-1/CTLA-4 inhibition has proven effective in several tumor types, improving on responses to single-agent ICB-therapy, and is now standard of care in malignant melanoma[27] and renal cell carcinoma[28].

In this phase II study, we demonstrate that the combination of nivolumab and ipilimumab is active in patients with R/M NPC, even after failing prior first-line chemotherapy. Given the need to identify potential biomarkers and understand the basis of tumor response, we analyze blood samples and paired biopsies (where feasible) obtained prior to treatment (termed pre-treatment) and at 4-6 weeks from the start of treatment (termed on-treatment).

## Results

### Patient characteristics, enrollment, and analysis
The trial schema is summarized in Fig. 1a. A total of 43 patients were enrolled between July 2017 and August 2019 across three sites. Three patients were excluded from efficacy analysis: one failed screening, one violated the inclusion criteria, and another patient withdrew consent (before completing 1 full cycle of combination treatment; Fig. 1b). Hence, forty patients were included in the analysis with a data

## Table 1 | Clinical characteristics of trial cohort

| | Number of patients (%) N = 40 |
|---|---|
| **Date enrolled into study** | 21 Jul 2017 to 22 Aug 2019 |
| **Date of treatment start for patients** | 31 July 2017 to 5 Sept 2019 |
| **Age at start of trial, years old** | |
| Median (IQR) | 53 (47.8, 61.1) |
| Range | 23-73 |
| **Gender** | |
| Male | 33 (82.5) |
| Female | 7 (17.5) |
| **Ethnic group** | |
| Chinese | 36 (90.0) |
| Malay | 3 (7.5) |
| Others | 1 (2.5) |
| **ECOG performance status at baseline** | |
| 0 | 12 (30.0) |
| 1 | 28 (70.0) |
| **Prior chemotherapy** | |
| No | 1 (2.5) |
| Yes | 39 (97.5) |
| **Prior radiotherapy** | |
| No | 13 (32.5) |
| Yes | 25 (62.5) |
| Unknown | 2 (5.0) |
| **Disease status** | |
| Metastatic | 34 (85.0) |
| Recurrent | 6 (15.0) |
| **Sites of metastases (n = 34)** | |
| Lung | 10 (29.4) |
| Liver | 12 (35.3) |
| Bone | 16 (47.1) |
| Adrenal | 1 (2.9) |
| Distant lymph node | 9 (26.5) |
| Neck | 1 (2.9) |
| Oropharynx | 1 (2.9) |
| Retropharyngeal soft tissue bid mediastinal lymph | 1 (2.9) |

## Table 2 | Summary of response and disease-control rates in trial cohort

| | Number of patients (%) N = 40 |
|---|---|
| **Best overall response (BOR)** | |
| Partial response (PR) | 15 (37.5) |
| Stable disease (SD) | 7 (17.5) |
| Progressive disease (PD) | 17 (42.5) |
| Not evaluable (NE) | 1[a] (2.5) |
| **BOR rate (95% CI)** | 38% (22.7% to 54.2%) |
| **Disease control rate, DCR (CR/PR/SD)** | |
| DCR (CR/PR/SD, regardless of SD duration) | 22 (55.0) |
| DCR, regardless of SD duration (95% CI) | 55% (38.5% to 70.7%) |

[a]Not estimable as only one tumor evaluation post-treatment.

55.7%). In the entire cohort of 40 patients (Phase II + expansion), a total of 15 patients experienced a BOR of PR (37.5%; 95% CI: 22.7% to 54.2%), and a disease control rate (DCR) of 55% (95% CI: 38.5% to 70.7%). Seven (17.5%) of 40 patients experienced stable disease (SD) and 17 (42.5%) experienced progressive disease (PD; Table 2). Responses were deep and durable (Fig. 2a, Supplementary Fig. 1). For the cohort of patients that responded (n = 15), the median time to response (TTR) was 3.1 months (95% CI: 2.5 to 3.7) and median duration of response (DOR) was 7.9 months (95% CI: 5.5 to not reached). With a median follow up of 26.2 months, 33 patients experienced disease progression and 27 patients passed away. Median PFS and OS were 5.3 (95% CI: 3.0 to 6.8) and 19.5 (95% CI: 13.1 to 23.9) months, respectively (Fig. 2b). Median time to progression (TTP) was 5.3 (95% CI: 3.0 to 6.8).

Thirty-four patients (85%) experienced any grade treatment-related adverse events (trAE; Supplementary Table 2), with the most common being maculo-papular rash (n = 16; 40%) and hypothyroidism (n = 11; 28%). In total, seven patients (18%) experienced trAEs of Grade 3 or higher. Four out of 40 patients (10%) experienced Grade 3 trAEs and two experienced Grade 4 trAEs (5%). One patient was recorded to have died of sudden death deemed possibly related to nivolumab and ipilimumab. Notably, of five patients who discontinued treatment due to severe adverse events (SAE), three continued to have prolonged tumor responses in the absence of any further treatment.

### Plasma circulating EBV-DNA correlates with response

Pre-treatment plasma cell-free EBV-DNA levels were available for the first 26 patients on study. Median pre-treatment levels were 6762 IU/mL (interquartile range: 883–30,039). We evaluated the ability of EBV-DNA to predict response to dual therapy using receiver operator characteristic (ROC) curves. These showed areas under the curve (AUC) of 64 (95% CI: 42.2–86.6) and 68.0 (95% CI: 40.8–95.2) for EBV-DNA when comparing PR vs SD/PD or PR vs PD, respectively (Fig. 3a). We identified an optimal threshold of 7800 IU/mL that dichotomized the cohort to 14 patients with low viral load (EBV$_{low}$) and 12 patients with high viral load (EBV$_{high}$; Supplementary Fig. 2a). Low viral load trended with better response to dual therapy: amongst EBV$_{low}$ patients, eight had confirmed PR (57.0%), compared with two patients with PR in the EBV$_{high}$ cohort (16.7%; OR = 0.16, 95% CI: 0.013–1.20, p = 0.051, Fisher's exact test). EBV$_{low}$ patients also had a superior median PFS of 6.8 months compared to 2.8 months in EBV$_{high}$ patients (p value 0.028 by log rank test) (Fig. 3b, Supplementary Table 3). Correlation to the depth of response and time on therapy also trended better in the EBV$_{low}$ cohort.

### Genomic analyses reveal the mutational landscape of the study cohort

A total of 22 fresh, immediate pre-treatment and 19 matched on-treatment biopsies, all with matched sequential blood samples were

cut-off date of 28 February 2021. Median follow up was 26.2 months (95% CI 22.5 to 31 months). The median age at the start of the trial was 53 years (range: 23 to 73 years). Majority (90.0%) were of Chinese ethnicity, and 33 patients (82.5%) were male. Thirty-nine patients (97.5%) had prior chemotherapy treatment before study entry (Table 1, Supplementary Table 1). All 40 patients received combination nivolumab and ipilimumab. As of data cut-off date, 38 patients had completed study treatment and two patients were still on study (Fig. 1b). Thirty-one patients discontinued treatment due to progressive disease, five patients discontinued due to adverse events and the remaining two due to investigator's decision and death, respectively. One patient continues to be on treatment with no dose modification. Nineteen patients experienced dose delay or omission of either drug.

### Efficacy and safety of the Phase 2 trial

In the stage one cohort of 15 patients, 7 (46.7%) reported best overall response (BOR) of partial response (PR), hence the trial proceeded to recruit another 11 patients into stage two. At the end of stage two of the Phase II trial, 9 out of 26 patients had a BOR of PR (35%; 95% CI: 17.2% to

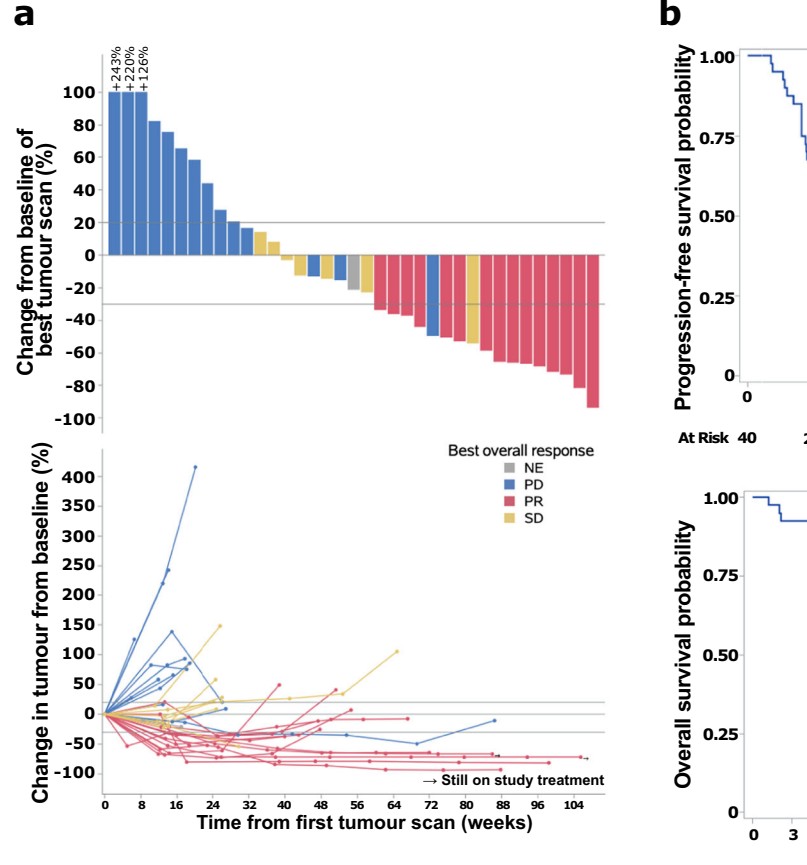

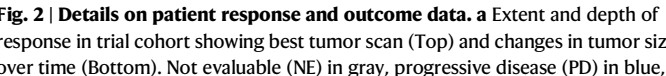

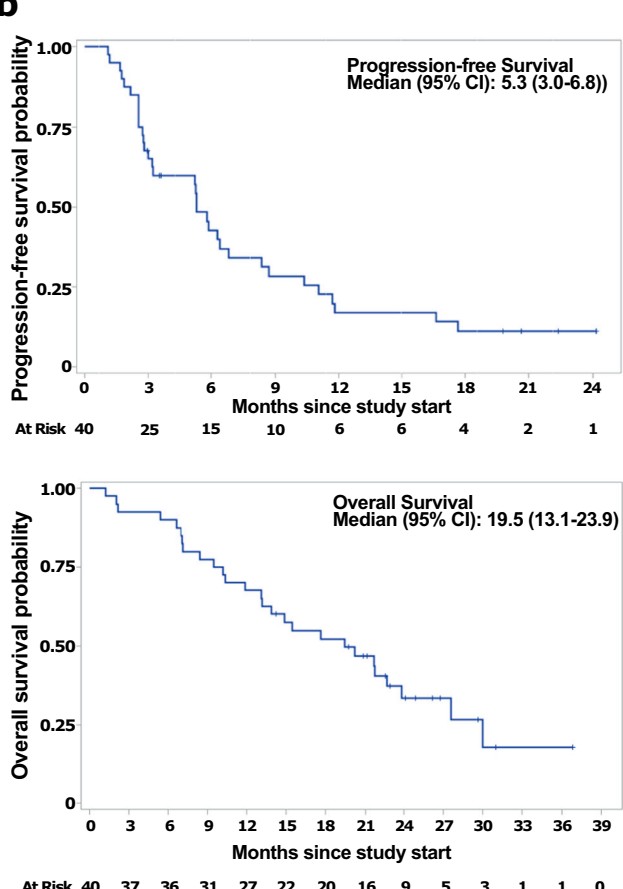

**Fig. 2 | Details on patient response and outcome data. a** Extent and depth of response in trial cohort showing best tumor scan (Top) and changes in tumor size over time (Bottom). Not evaluable (NE) in gray, progressive disease (PD) in blue, partial response (PR) in red and stable disease (SD) in yellow. *n* = 40. **b** Progression-free (Top) and overall (Bottom) survivals of trial cohort. *n* = 40. Source data are provided as a Source Data file.

available for immunophenotyping and biomarker analysis (Fig. 3c, Supplementary Table 4). These samples were collected as part of a planned analyses of the genome, transcriptome and histopathology although the exact methodology were not pre-specified as it depended on the eventual quality of samples and assays available. For these analyses, response was based on specific response at the site where the tissue was obtained, rather than BOR, and as such we included a further two patients who did not meet the timing cutoff for outcomes (033 and 040). To determine the genomic landscape, identify active mutational signatures, and explore the utility of tumor mutation burden (TMB) as a biomarker of response, whole-exome sequencing was performed on available tissue and matched blood normal (*n* = 20). Genomic analysis showed that the mutational landscape of these tumors was generally bland with a median TMB of 0.75 mut/Mb (range: 0.04–14.56), corroborating previous reports[3,24,29–31] (Fig. 3d, Supplementary Fig. 2b). The most frequently mutated genes included *TP53*, *FAM135B*, *COL3A1* and *EP300* (Fig. 3d, Supplementary Fig. 2c). The most common mutational signatures were Cosmic signatures SBS5 and SBS40, both of unknown etiology (Fig. 3e). Signatures associated with cytosine deamination were also frequently detected, with spontaneous deamination of 5-methylcytosine (SBS1) detected in nine tumors and APOBEC signatures (SBS2, SBS13) enriched in six. There was no correlation between TMB or any specific mutational signature with treatment response (Supplementary Fig. 2d–e). However, the four patients with highest TMB (>2 mut/Mb) achieved PR, with one (TMB of 5.02 mut/Mb) demonstrating particularly durable clinical response that lasted for 12 months after start of therapy.

## Expression profiling shows distinct gene and pathway differences between responders and non-responders after dual therapy

Expression profiling using the Nanostring IO360 panel was performed on all 22 pre-treatment and matched 19 on-treatment samples (Fig. 3c). This panel comprises a curated list of 750 genes, representing genes expressed by tumor cells and the immune compartment. We investigated both pre-treatment and on-treatment Nanostring gene expression profiles for associations with response. Expression profiling of pre-treatment samples could not be resolved by treatment response, likely reflecting the heterogeneity consequent to disease presentation, biopsy site and prior therapy (Supplementary Fig. 3a–c). Previously reported signatures such as an IFNγ signature[32] and a cytotoxic T-cell signature (tGE8)[33] did not correlate with response when applied to pre-treatment biopsies (Supplementary Fig. 3d–e). Additionally, *CD274* (PD-L1) gene expression failed to differentiate responders from non-responders as expected (Supplementary Fig. 3f), together suggesting the pre-treatment profiles did not contain robust biomarkers of response.

In contrast, analysis of on-treatment samples showed remarkable expression differences between responders (PR, *n* = 8) and non-responders (PD, *n* = 5). Differential expression analysis of PR vs PD samples identified a panel of 158 genes of interest (DESeq2 *p*-value < 0.05; Fig. 4a, b, Supplementary Data 1). These genes are involved in a number of critical pathways driven primarily by an increase in adaptive immune response: cytotoxic T-cell activation, response, signaling, and B-cell mediated activation were upregulated in responders (Fig. 4c). In comparison, non-responders showed upregulation of DNA damage

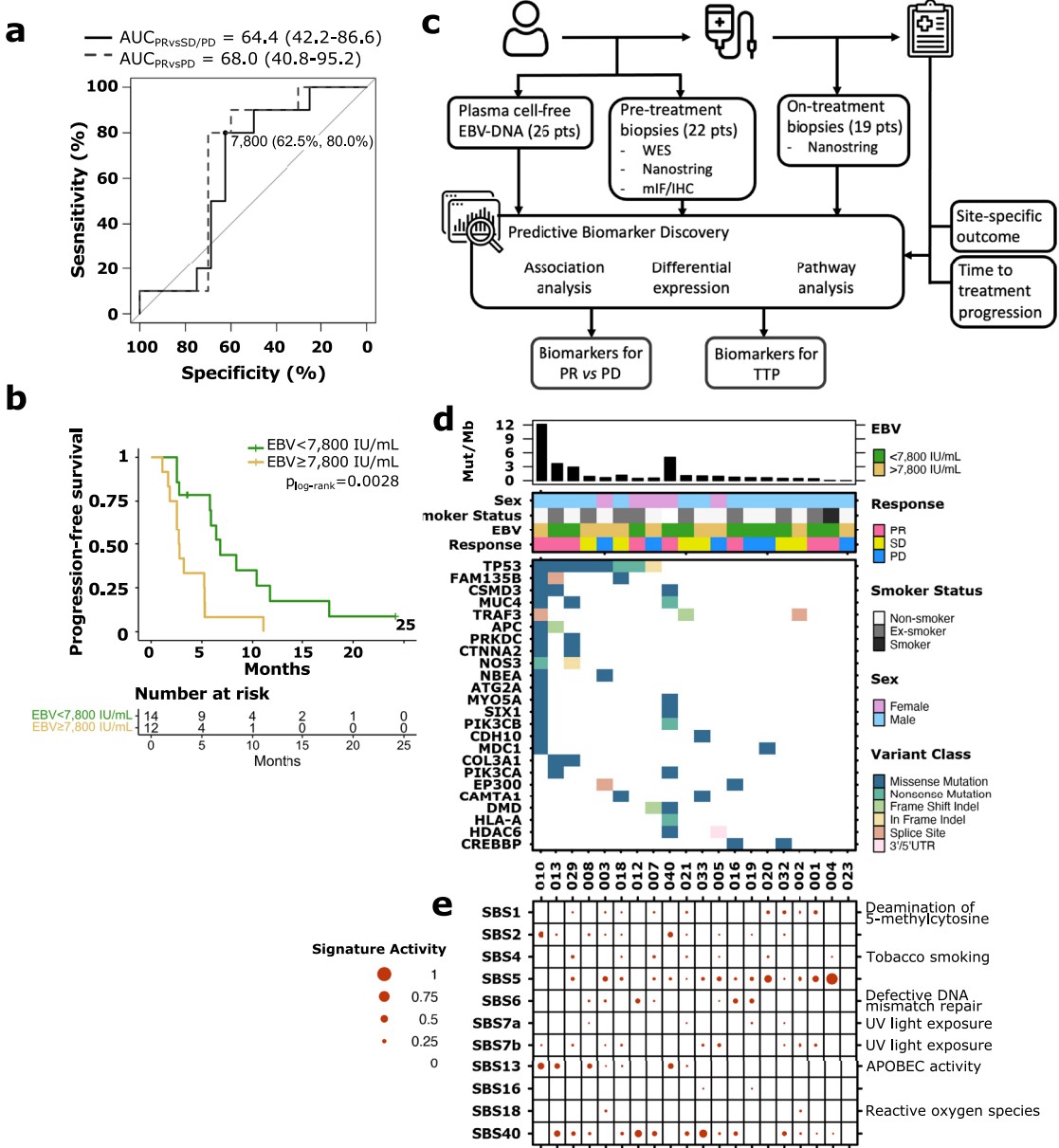

**Fig. 3 | Pre-treatment plasma circulating EBV-DNA levels and mutational landscape of tumors by whole-exome sequencing. a** Predictive ability of plasma circulating EBV-DNA levels for partial response to dual therapy (*n* = 26). **b** Progression-free survival for EBV-high (yellow) vs EBV-low (green) patients with two-sided log-rank *p*-value shown. **c** Summary of molecular assays performed and biomarker discovery workflow. Icons made by Freepik and monkik on www.flaticon.com. **d** Tumor co-mutation plot showing top recurrently mutated cancer genes in cohort. Tumor mutation burden and clinical covariates are shown. **e** Mutational signatures based on COSMIC signatures. Size of dot correlates with the strength of signature activity in each sample. *n* = 20, biologically independent samples.

pathways and the innate immunity response including viral response and type 1 interferon signaling. These transcriptomic changes all occurred within four weeks of initiation of CTLA-4 and PD-1 blockade, prior to any overt clinical response.

We expanded these analyses to incorporate the patient cohort with SD by performing Pearson correlation analysis on gene expression with TTP (*n* = 18 samples; Fig. 4d, Supplementary Data 2). This revealed a number of genes whose expression positively correlated with TTP including those involved in the adaptive immune response (*PDCD1*, *HAVCR2*, *CCL5*, *CD244*), compared to genes involved in proliferation/DNA damage (*MSH6*, *LDHB*, *BIRC5*, *EGFR*), which were negatively correlated.

The pre- and on-treatment sample pairing presented a unique opportunity to investigate dynamic changes in response to therapy.

We examined the 158 on-treatment genes of interest in pre-treatment expression profiling and found no association with response (Supplementary Fig. 4a). Next, we compared the differential expression of the on-treatment genes of interest between PD vs PR separately in pre-treatment and on-treatment samples; that is, we used Mann-Whitney U-tests to compare the expression of each gene in on-treatment PD vs on-treatment PR and pre-treatment PD vs pre-treatment PR, separately. We found that while there were strong associations in on-treatment samples, the effects in pre-treatment samples were weak and not statistically significant (Supplementary Fig. 4b, Supplementary Data 3). Returning to pre-treatment samples, we identified seven differentially expressed genes between PD vs PR patients (Supplementary Fig. 4c) and similarly assessed the pre-treatment genes of interest in on-treatment samples (Supplementary Fig. 4d). The resolution of PD vs

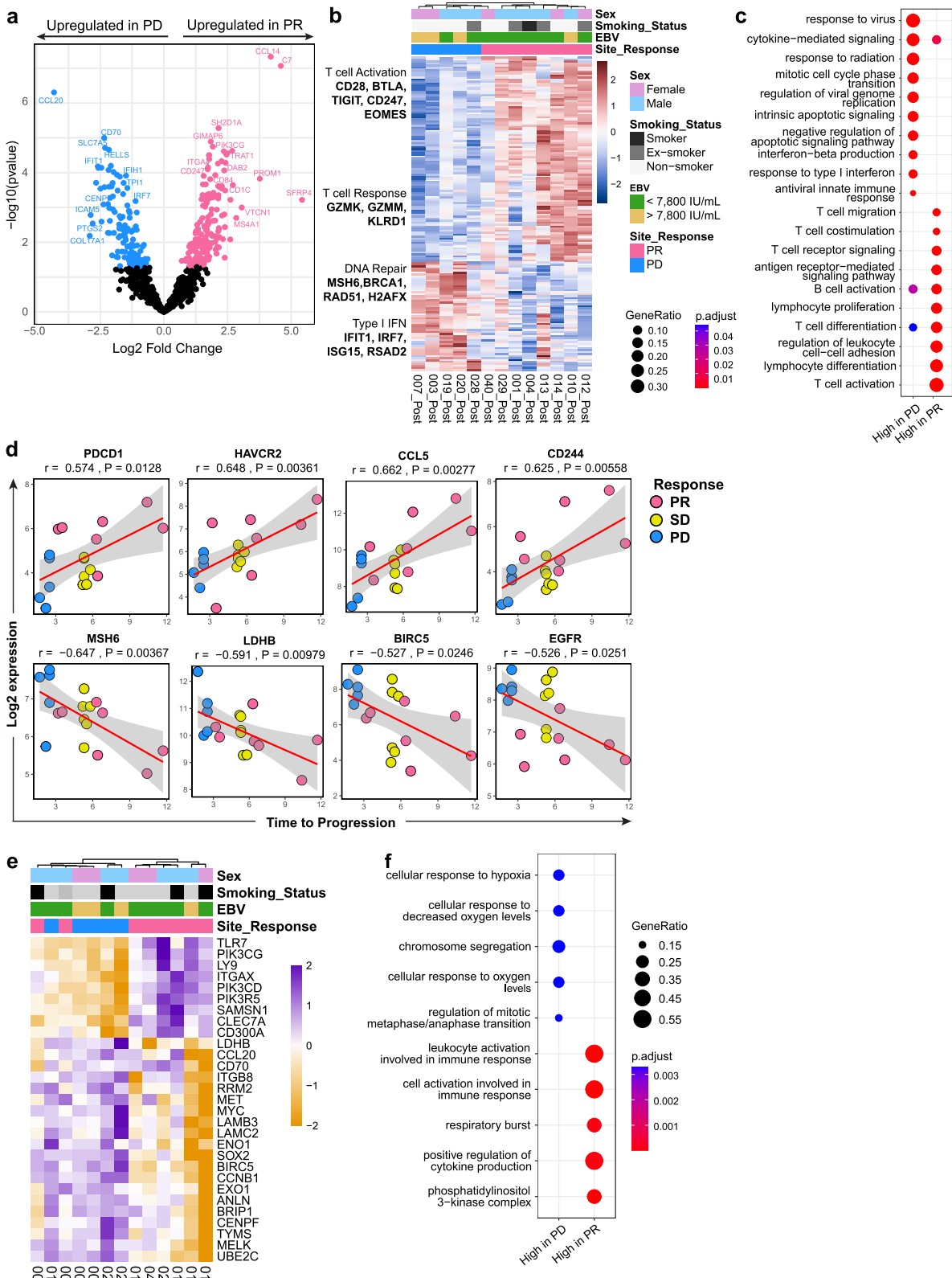

**Fig. 4 | Gene expression analysis of tumor biopsy samples pre- and on-treatment. a–c** Results from differential expression analysis of on-treatment Nanostring IO360 data for PR *vs* PD outcome with (**a**) volcano plot and (**b**) heatmap highlighting key genes, and (**c**) pathway enrichment analysis of significantly differentially expressed genes. $n_{PR} = 8$, $n_{PD} = 5$. **d** Time to progression is positively correlated with markers of adaptive immune response (top) and negatively correlated with markers of proliferation and DNA damage response (bottom; *r* and two-sided *p* values determined using Pearson correlation statistical analysis). Each dot represents an individual sample. $n = 18$ ($n_{PR}=7$, $n_{SD} = 6$, $n_{PD} = 5$). **e–f** Gene expression changes associated with the interaction between treatment status (pre- vs on-treatment) and outcome (PR *vs* PD) and (**f**) their enriched pathways. $n_{PR} = 8$, $n_{PD} = 5$, biologically independent samples. **a**, **b**, **e** Statistical analyses for differential expression were performed using Wald-test, not corrected for multiple testing.

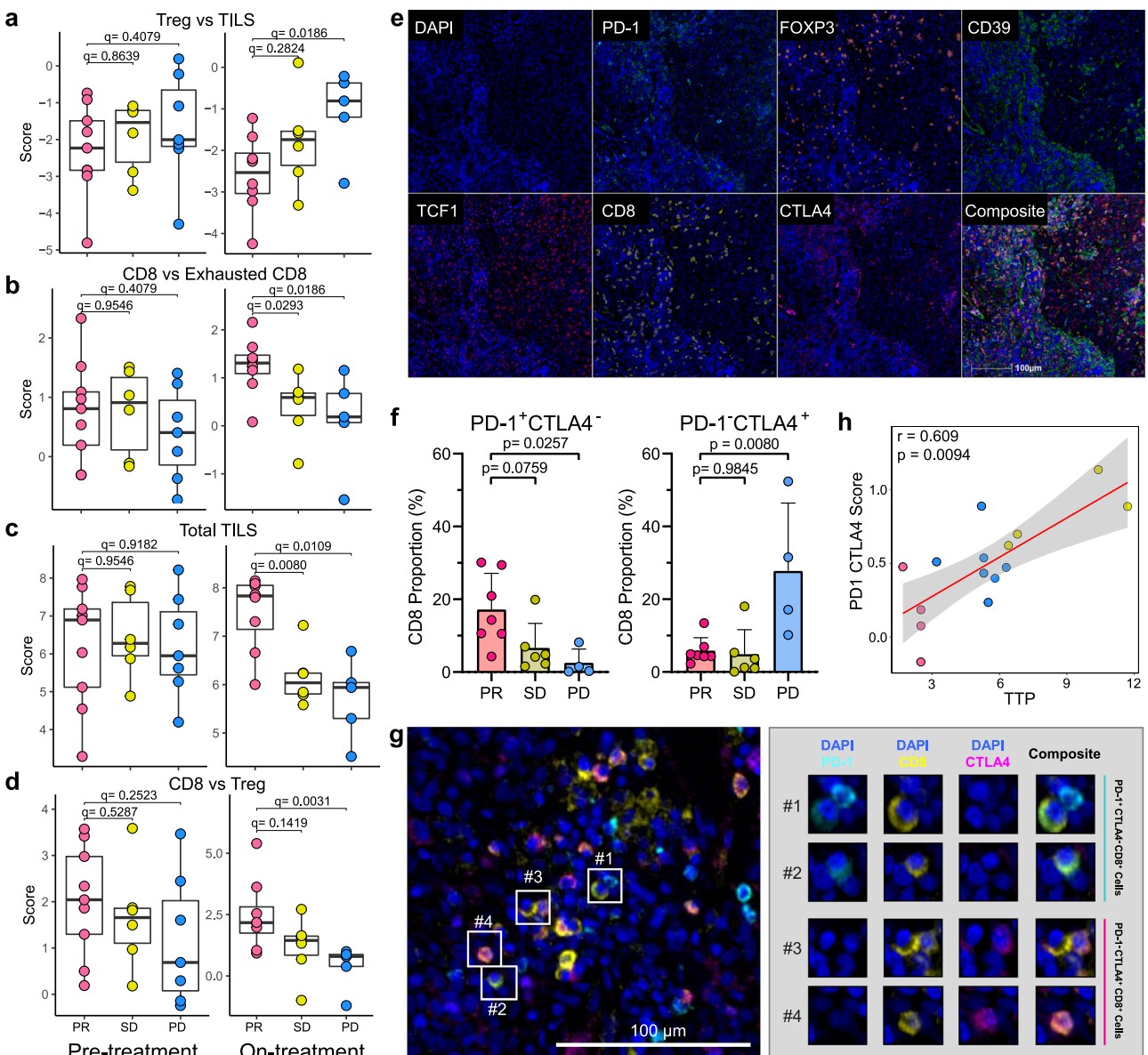

**Fig. 5 | Cell type profiling by Nanostring and multiplex IHC/IF staining.** Cell type profiling score (as determined using Nanostring IO360) of (**a**) Treg vs TILS, (**b**) CD8 vs Exhausted CD8, (**c**) Total TILS, and (**d**) CD8 vs Treg by outcome compared between pre- (left) and on-treatment (right) samples as indicated at the bottom of the graph. Pre-treatment = 22 ($n_{PR}$ = 8, $n_{SD}$ = 6, $n_{PD}$ = 7), On-treatment=19 ($n_{PR}$ = 8, $n_{SD}$ = 6, $n_{PD}$ = 5), biologically independent; two-sided Mann−Whitney $U$-test, adjusted for multiple comparisons, boxplots are shown with the boxes indicating quartiles with median at middle and the whiskers drawn at the lowest and highest points within 1.5 interquartile range of the lower and upper quartiles, respectively. Each dot represents an individual sample. (**e**) Representative images of 17 patients stained for DAPI (blue), PD-1 (cyan), FOXP3 (orange), CD39 (green), TCF1 (red), CD8 (yellow), CTLA-4 (magenta; 200x magnification), performed in 2 independent experiments. **f** Quantification of PD-1+ CTLA-4- CD8+ and PD-1- CTLA-4+ CD8+ cells as proportions of total CD8 cells. Tukey's test, adjusted for multiple comparison. Data are presented as mean with SD. $n_{PR}$ = 7, $n_{SD}$ = 6, $n_{PD}$ = 4. Each dot represents an individual sample. **g** Representative image of combined CD8 subpopulations described in (**f**) expressing either PD-1 or CTLA-4. **h** Correlation between PD-1/CTLA-4 score with TTP. $n$ = 17, Spearman correlation $r$ = 0.609, $p$ = 0.0094. Each dot represents an individual sample. Source data are provided as a Source Data file.

PR was moderate though not as strong as using the on-treatment genes of interest (Supplementary Fig. 4e) and the differences in expression for 5/7 pre-treatment genes of interest were not statistically significant in on-treatment samples (Supplementary Data 3). However, the directions of differences were preserved and these genes may represent a condensed gene signature for discriminating PD vs PR patients.

Finally, we modeled the transcriptomic differences between pre-treatment PR vs PD and on-treatment PR vs PD including an interaction term, and identified 29 associated genes (Fig. 4e, Supplementary Data 4). Pathway enrichment analysis of these genes showed consistent immune involvement and downregulation in tumor cell proliferation and hypoxia pathways (Fig. 4f). These results indicate that in

responders, despite the vastly different initial profiles in each tumor, there is a remarkable convergence in the immune microenvironment soon after dual ICB therapy prior to detectable clinical response.

## Multiplex immunohistochemistry shows immune subpopulations that correlate with treatment response

Most of the currently available models suggest that immune checkpoint blockade functions by reinvigoration of existing and recruitment of circulating CD8 populations. In addition, CTLA-4-blockade is also known to downregulate Tregs[34]. Cell type signature analysis from our Nanostring IO360 data showed evidence of these phenomena in our samples, with lower Treg populations, lower proportions of CD8 vs

exhausted CD8, more tumor infiltrating lymphocytes, and more CD8 vs Treg proportions in on-treatment samples of partial responders compared with non-responders (Fig. 5a–d). We didn't find any significant difference in these cell proportions in pre-treatment samples. To further investigate immune subpopulations, we performed multiplex immunohistochemistry/immunofluorescence (mIHC/IF) staining of pre-treatment biopsies ($n = 17$) to experimentally identify the different T-cell lineages present in these tumors prior to treatment. Markers used include CD8, FOXP3, PD-1, CTLA-4, CD39, and TCF1 (Fig. 5e, Supplementary Table 5). There were no correlations between the major lineages defined by CD8, CD4 or FOXP3, nor their relative ratios, with response in concordance with the transcriptomic data (Supplementary Fig. 5a). Similarly, there were no correlations with specific CD39 or TCF1 fractions.

However, there was a correlation between pre-treatment immune subpopulations expressing the two targets of the drugs used in this study (PD-1 and CTLA-4) with response. Specifically, the PD-1+CTLA-4−CD8+ immune cell proportion was significantly higher in responders, while the PD-1−CTLA-4+CD8+ population was more commonly observed in non-responders (PD vs PR; Fig. 5e–g). These two subpopulations appeared to be independent of EBV titre (Supplementary Fig. 5b) and each other (Supplementary Fig. 5c). Combining these two sets of data into a composite score showed good correlation between the proportions of these two specific subpopulations with TTP (Spearman correlation $r = 0.609$, $p = 0.0094$; Fig. 5h). These results suggest an intriguing notion that the presence of distinct CD8 subpopulations, mutually expressing the two different checkpoint inhibitors exclusively, could portend the response to this combination of PD-1/CTLA-4 -blockade, and hence could be used as a biomarker predictive of response.

## Discussion

This study demonstrates that the combination of nivolumab and ipilimumab is active in R/M NPC even after prior exposure to first line combination chemotherapy. Within the limitations of cross-trial comparison with reported single agent PD-1 studies, there is increased activity evidenced by higher BOR rate, longer median PFS, and median duration of responses[35]. Responses occurred early within 3 months and there was only one case of pseudo-progression where response was seen after 6 months. Of note, 3 patients appeared to hyperprogress (Fig. 2a). However only one patient had sufficient tissue for biomarker analysis, and we were unable to elucidate further possible mechanisms for this phenomenon. In contrast to reported single agent studies, where one or two complete responses were reported, we did not see any in this study. This is likely a function of sample size, although a difference in the biological response to dual immunotherapy in NPC cannot be excluded.

Importantly, the median PFS and median OS seen in the study were 5.3 moths and 19.5 months, respectively, with a disease control rate of 55% and a proportion of patients had achieved durable responses to the combination treatment. This compares favorably to reported median PFS achieved with second-line chemotherapy as detailed by Prawira et al. in their systematic review of patients treated with second line chemotherapy, where the median PFS was 5.4 months (95% CI 3.8–7.0), and is superior to the median OS reported as only 11.5 months (95% CI 10.1–12.9)[11]. However, in reported studies of similar PD-1/CTLA-4-blockade combinations in head and neck squamous cancer, this was not superior to standard EXTREME combination chemotherapy[36,37], suggesting possible limitations of a checkpoint inhibitor only strategy. In a relatively more chemotherapy-sensitive tumor like NPC, a formal comparison of chemotherapy and dual checkpoint inhibitors compared to chemotherapy and PD-1 inhibitors in R/M NPC would be required.

The safety profile of the combination is similar to reported rates in other studies of this combination in solid tumors[38,39]. Overall, the toxicities were manageable and grade 3/4 toxicities observed were readily reversible with appropriate steroid intervention and medical support, although not unexpectedly the rate is higher than that with monotherapy checkpoint inhibitors[40]. It is also noteworthy that some patients who had trAE requiring discontinuation showed durable responses even in the absence of next line treatment, consistent with recent data presented in non-small cell lung cancer[41]. This acceptable toxicity profile and clinical efficacy may allow this dual blockade to be further explored in combination with chemotherapy or radiotherapy, and as maintenance in the R/M treatment naïve metastatic or locally advanced disease setting.

Conventional PD-L1 expression does not correlate well with response to PD-1/PD-L1 blockade in NPC[18,20], unlike in non-small cell lung[42] or head and neck squamous cell cancers[17]. As this study was not powered to show this association, we did not proceed to do PD-L1 IHC given the limited tissue available from the biopsies which were being subjected to other biomarker analyses. *CD274* (PD-L1) mRNA expression on Nanostring was done but it did not correspond to treatment outcomes (Supplementary Fig. 3f). Quantifying pre-treatment circulating EBV-DNA shows a trend to increased activity of this combination in patients with low plasma titers and presents a potential opportunity to select/stratify patients for treatment using an EBV-DNA-based biomarker. A possible explanation for this divergence in response is that high EBV and tumor load could represent a diverse immunosuppressive tumor microenvironment with high cellular proliferation and cell cycling, and hence a more aggressive subset of disease with poorer prognosis[43,44]. Loss of function of EBV-specific T-cells within NPC[45] has also been observed in this setting, possibly leading to reduced tumor-directed efficacy. Pre-treatment analyses of T-cell subset/subpopulations provide a method of enriching for responders with identification of specific PD-1 and CTLA-4 expressing CD8 subsets that associate with response, with intriguing implications on the interactions between these immune subsets, EBV-positive tumor cells, and drugs targeting PD-1 and CTLA-4. Similar upregulation of PD-1 on CD8 T-cells has been previously described with single-cell RNA sequencing of NPC[46]. Thus, the exhaustion phenotype demonstrated here on mIF/IHC may correspondingly represent a clinically applicable pre-treatment biomarker for response to dual immunotherapy. Indeed, the current study has now been extended further to validate and confirm these observations in a larger patient cohort and establish a correlation between EBV burden and response.

Deep tumor immunophenotyping in this study demonstrated the importance of expression profiling in NPC, especially given the paucity of existing biomarkers for checkpoint inhibitor response; TMB and PD-L1 expression show little or no association with response in NPC. This could be a function of tumor heterogeneity, but it could also suggest lack of PD-L1 dependence in NPC[47,48]. Given the bland genomic landscape, TMB in NPC is not high and this further limits the application of TMB as a single predictive biomarker in NPC as shown here and other studies[24]. While expression profiling of pre-treatment samples similarly highlighted the heterogeneity of these tumors, on-treatment gene expression showed remarkable convergence of the immune response across responders, with consistent activation of adaptive immunity and cytotoxic T-cell response. The increase in B-cell activation could also indicate the presence of tertiary lymphocyte structures that have been identified as a potential biomarker for this same drug combination in bladder cancer[49]. In contrast, expression profiling of on-treatment samples in non-responders have identified targets and aberrant pathways that could putatively identify alternative targets. These include other immune checkpoint inhibitors (e.g. TIGIT, LAG3, TIM3) or alternative pathways critical to immunomodulation (e.g. hypoxia, DNA damage activation). These results warrant future studies to investigate different combinations of immune blockade, in addition to validating these biomarkers in independent cohorts.

We acknowledge that the non-randomized nature of the study lacks a direct control arm of standardized chemotherapy for comparison of activity and is one of the limitations of this study. The number of patients and tissues available for analysis also makes definitive biomarker correlations limited to generating hypotheses for further validation and testing in larger cohorts. Finally, given the introduction of PD-1 inhibitors into the first line treatment space, we do not know the activity of this combination in patients who had prior exposure to checkpoint inhibitor.

Despite the absence of formal comparative studies, combination nivolumab and ipilimumab presents a promising chemotherapy-free alternative to post-first line salvage chemotherapy in R/M NPC. It also merits further trials on how best to incorporate this into fast evolving standards of care for NPC. Future studies of combinations which incorporate pre-treatment selection and stratification, as well as on-treatment molecular signature-based adaptive designs will be useful to define additional cohorts for enrichment and study.

## Methods

### Trial Oversight

The study was designed and conducted in compliance with ICH Good Clinical Practice guidelines and ethical principles described in the Declaration of Helsinki, regarding the use of human subjects in clinical trials. The study was approved by the respective Institutional Review Boards (Singapore Health Services Institutional Review Board, National Healthcare Group Domain Specific Review Board, and National Taiwan University Hospital Research Ethics Committee), and all patients provided written informed consent prior to enrollment and starting any study procedures, and the study followed CONSORT reporting guidelines.

### Patients

Patients who met the study eligibility criteria and provided written informed consent were recruited from three academic centers in Taiwan and Singapore. No selection bias existed in patient recruitment for this single-arm Phase II trial. They were eligible if they had recurrent/metastatic (R/M) undifferentiated NPC with detectable plasma EBV-DNA at study entry. They had to be at least 20 years of age at study entry, have measurable disease per RECIST v1.1 criteria, and of good ECOG PS 0/1. They could not have received more than one line of prior palliative chemotherapy. Patients who progressed/relapsed within 6 months of definitive chemoradiation for locally advanced disease were considered to have chemotherapy-resistant disease, and patients who were not fit for platinum-based therapy disease were eligible. Patients did not receive financial compensation for participation in the study.

### Trial design and treatment

This was an investigator-initiated single-arm phase II study of nivolumab and ipilimumab in R/M NPC. The study was registered in clinicaltrials.gov on 31 Mar 2017 (https://clinicaltrials.gov/ct2/show/NCT03097939). This study was designed using Simon's minimax two-stage design to investigate if the best objective response was at least 45% with a no-interest BOR rate of 25%. At 80% power and 10% significance level, 15 patients were to be recruited into the first stage. If at least 4 patients experienced best overall response (BOR) (composite of complete response [CR] and partial response [PR]), the study would proceed to the second stage. Eleven more patients were to be recruited to the second stage and if at least 10 of 26 patients experienced a BOR, the treatment combination would be considered worthy of further testing in a Phase III setting. Per protocol, an additional 14 patients were recruited to provide additional precision for the clinical efficacy and safety estimates.

Patients were treated with intravenous nivolumab 3 mg/kg every 2 weeks and ipilimumab 1 mg/kg every 6 weeks. One cycle of treatment was 6 weeks. The schedule and doses of this combination were adopted from initial phase I studies done in solid tumors and confirmed in expanded cohorts in lung cancer[32,33]. A dose given more than 3 days after the intended dose date was considered a delay. Both drugs were continued until disease progression, unacceptable toxicity, withdrawal of consent, or study end, whichever occurred first. Clinic visits and physical assessments were done every 6 weeks, tumor assessments were done by investigators per RECIST v1.1 every 12 weeks, and responses were confirmed with follow-up imaging within 6 weeks. EBV-DNA load was measured per established institutional protocols at baseline and at every cycle. All results are reported in IU/ml. Tumor biopsies and matched blood for biomarker analyses were obtained at baseline and on-treatment within 4-6 weeks of starting treatment, where available. The first patient first visit was on 21 July 2017 and the last patient first visit was on 22 August 2019.

### Endpoints

The primary efficacy endpoint was BOR, defined by the best response (composite of CR and PR) by RECIST v1.1 recorded from the start of study until disease progression/recurrence of R/M NPC to combination checkpoint inhibition, and dependent on the achievement of both measurement and confirmation scan. This outcome was decided prior to data collection and formed the basis for the sample size estimate for the study. Clinical benefit was defined as a best response of CR, PR or stable disease (SD). BOR rate of CR/PR and clinical benefit rate were reported with corresponding 95% confidence intervals (CI) estimated using the Clopper-Pearson method.

Secondary endpoints included clinical benefit rate (CR/PR/SD), progression-free survival (PFS), overall survival (OS), duration of response (DOR), time to progression (TTP), and frequency of adverse events. DOR was defined as the time from first assessment of CR or PR until the first date that progressive disease (or recurrent disease for patients who experienced CR) or death was objectively documented. TTP was defined as time from study entry until the first date that PD was objectively documented. PFS was defined as time from study entry until objective tumor progression, or death from any cause, whichever occurs first. Patients who were alive or did not have an assessment of PD were censored at the date of last tumor assessment. Patients who discontinue treatment were censored at their last tumor assessment date. OS is defined as time from study entry until death from any cause. Patients who were alive were censored at their date of last follow-up. For each time-to-event endpoint (TTP, PFS and OS), the survival distributions were estimated using the Kaplan–Meier product-limit method. The median time and corresponding 95% confidence interval were estimated using the Brookmeyer and Crowley method.

Adverse events were assessed at every visit. Severity of adverse events were graded according to the NCI Common Terminology Criteria for Adverse Events (CTCAE) version 4.0. If a patient experienced more than one incidence of an AE during the trial, the worst grade experienced by the patient was reported.

### Protocol deviations

At the end of stage 2 of the Simon 2-stage design, 9 of 26 (35%) patients met the BOR criteria. The preplanned rule in the original protocol was written in 2017, prior to any available estimation of the average BOR seen in NPC for PD-1 monotherapy. The observed BOR in the first 26 patients of 35% was well above the no-interest BOR of 25% set in the protocol, and the differential response rates and outcomes in correlation to baseline plasma EBV-DNA levels in the initial 26 patients were deemed too significant to ignore by the study team, and supported by the institutional review board. Hence, the study was continued and completed accrual of the pre-planned sample size of 40 patients, to better assess efficacy and toxicity, and reported in full here with the consent of all authors. Based on the preliminary results of the association of plasma EBV-DNA with response in the first 26 eligible patients, the trial was further expanded to recruit more patients to test

the association of plasma EBV-DNA with response. At the point of time of time of this report, this expansion cohort has closed recruitment and follow-up is ongoing.

Planned iRECIST reporting was not done due to resource constraints resulting from the SARS-COV-2 pandemic. Tissue samples were collected as part of a planned analyses of the genome, transcriptome and histopathology (see below for details) although the exact methodology was not pre-specified as it depended on the eventual quality of samples and assays available.

## DNA and RNA extraction
DNA and RNA were extracted simultaneously from frozen tissue biopsies using All Prep RNA and DNA Mini Kit (Qiagen GmbH, Hilden, Germany) according to manufacturer's protocol. DNA and RNA were stored at −20°C and −80°C respectively until use.

## Whole exome sequencing analysis
Genomic DNA was processed using Agilent SureSelect Human All Exon V6 (Agilent Technologies, Santa Clara, CA, United States) as described in the manufacturer's protocol[50], and then subjected to paired-end 150 bp sequencing using NovaSeq 6000 (Illumina, San Diego, CA, United States). The sequencing depth for tissue samples was 200X and for blood samples was 100X. Sequencing reads were trimmed using Trim Galore and processed according to The Genome Analysis Toolkit (GATK) best practices (https://software.broadinstitute.org/gatk/best-practices/workflow?id=11165). Briefly, reads were aligned against human reference build GRCh38 using BWA mem (v0.7.17) and SAM files were converted to BAM files using samtools (v1.7). MarkDuplicatesSpark from GATK (v4.1.8.0) was used to mark duplicates and sort BAM files, and BaseRecalibrator and ApplyBQSR were used to perform base quality score recalibration. Somatic single nucleotide variants were called using a consensus of three variant callers run with default parameters: Mutect2 (GATK v4.1.8.0), Strelka2 (v2.9.10) and Lancet (v1.1.0). SNV calls from each caller were filtered for common germline, low confidence, and sequencing artifact variants. We took somatic variants called by at least two algorithms. These consensus calls formed the basis for recurrent SNV and mutational signatures identification. We used deconstructSigs (v1.8.0) to identify mutational signatures for each tumor.

## Gene expression profiling using Nanostring nCounter
A total of 125 ng of RNA from each sample was subjected to PanCancer IO 360 Gene Expression Panel (Catalog No. XT-CSO-IO360-12) using nCounter MAX Analysis System (NanoString Technologies, Inc., Seattle, WA, USA) following the manufacturer's protocol. Raw expression data was normalized and analyzed using nSolver Advanced Analysis feature in nSolver analysis software (v4.0; NanoString Technologies, Inc.) to obtain cell type scores. Raw expression data was normalized using RUVSeq (v1.24.0) for differential expression analysis in R.

Differential expression analysis comparing PR vs PD was iteratively performed using DESeq2 R package (v1.34.0) excluding one sample at a time; this resulted in 14 sets of differentially expressed genes. Overlapping genes were considered as a final and robust set of differentially expressed genes. DE analysis comparing pre- and on-treatment samples for PR vs PD was also performed using DESeq2, controlling for individual patients. DE genes were then investigated using pathway analysis with clusterProfiler (v4.1.1).

Pearson correlation analysis was performed by comparing log2 normalized expression with time to progression. To evaluate previously published gene expression profile scores, we calculated the weighted score from 18 IFNγ related genes as described by Ayer et al[32]. Cytotoxic T-cell transcriptional signature (tGE8 score) comprising of 8 genes, namely *IFNG*, *CXCL9*, *CXCL10*, *CD8A*, *GZMA*, *GZMB*, *PRF1* and *TBX21*, were calculated using GSVA (v1.41.1). *CD274* (PD-L1) mRNA expression were plotted using log2 normalized expression. Statistical analyses were performed using Kruskal-Wallis test.

## Multiplex immunohistochemistry/immunofluorescence (mIHC/IF)
mIHC/IF was performed on FFPE biopsy tissue using Opal Multiplex fIHC kit (Akoya Biosciences, Menlo Park, CA, USA)[51]. In brief, 4 um FFPE tissue sections were semi-automatically processed using Leica Bond Max autostainer (Leica Biosystems, Wetzlar, Germany). The slides were stained using 6 primary antibodies (CD8, FOXP3, PD-1, CTLA-4, TCF1 and CD39; summarized in Supplementary Table 5), polymeric HRP-conjugated secondary antibodies from BOND Polymer Refine Detection kit (Cat DS9800; Leica Biosystems, Newcastle, UK), and Opal fluorophore-conjugated tyramide signal amplification (1:100 dilution; Cat #NEL797001KT; Akoya Biosciences, Marlborough, MA, USA). Mounted slides were scanned using Vectra 3 pathology imaging system microscope (Akoya Biosciences). Images were analyzed using inform software (v2.4.6; Akoya Biosciences). Statistical analysis and visualization were performed using GraphPad Prism (v8.0.0; GraphPad Software, Inc., San Diego, CA, USA).

## Statistical analysis
The primary endpoint of BOR rate was summarized using descriptive statistics with corresponding 95% confidence intervals estimated using the Clopper–Pearson method. Patient demographics and clinical characteristics at study were summarized as frequency and percentage, and continuous variables were summarized as median with interquartile range and/or range. PFS and OS were estimated using the Kaplan–Meier product-limit method. The median time and corresponding 95% confidence interval were estimated using the Brookmeyer and Crowley method. The trAEs were summarized as frequency and percentage.

Exploratory analyses were performed for the patients in the Phase II trial to investigate the association of EBV-DNA load with BOR and PFS. EBV-DNA load was measured per established institutional protocols, and pre-treatment EBV-DNA load of 7800 IU/ml was used to discriminate $EBV_{low}$ from $EBV_{high}$. Comparison of BOR rate and PFS between EBV groups were performed using the Fisher Exact test and log-rank test, respectively.

All eligible patients who received at least one dose of the combination drugs were included in the analyses. Two-sided *P*-values were reported for all analyses and *P*-values less than 0.05 were considered statistically significant.

All statistical analyses and visualizations were performed using SAS version 9.4 (SAS Institute Inc., Cary, NC), GraphPad Prism (v8.0.0; GraphPad Software, Inc., San Diego, CA, USA), and R version 4.1 with packages maftools (v2.8.05), survival (v3.2-13), survminer (v0.4.9), pROC (v1.18.0), BoutrosLab.plotting.general (v6.0.3), pheatmap (v1.0.12), and ggplot2 (v3.3.5). Icons in workflow diagrams were created by Freepik and monkik on www.flaticon.com. Hospital icons created by Freepik–Flaticon (https://www.flaticon.com/free-icons/hospital); Analysis icons created by monkik–Flaticon (https://www.flaticon.com/free-icons/analysis).

## Reporting summary
Further information on research design is available in the Nature Portfolio Reporting Summary linked to this article.

# Data availability
Data availability is subject to local rules and regulations. Patient data from a clinical trial is subject to patient confidentiality. Subjects did not provide consent for their DNA or clinical data to be made publicly available. However, requests to access clinical and sequencing trial data will be considered case by case. Information that may be considered for disclosure upon request includes de-identified participant clinical data, study protocol and statistical analysis plan. Given the restrictions posed by patient consent and institutional review boards, the raw WES data cannot been deposited in a public repository but

processed data can be made available upon request. Requests for data should be made to the corresponding authors together with a detailed study plan and a commitment not to use the data and its derivatives for commercial purposes. The proposal will require approvals by the respective institutional review boards and the principal investigators. Requesting researchers will be required to sign a data access agreement with the relevant parties. The raw Nanostring data is available in the GEO database under accession code GSE224450. The remaining data are available in the Article, Supplementary Information, or Source data file. Source data are provided with this paper.

## Code availability
No custom algorithms or software were developed or used in this study. Analysis scripts are available on GitHub (https://github.com/nccsCancerTherapeuticsLab/NPC_IpiNivoTrialAnalysis). All software and algorithms are publicly available and are listed in the Methods section.

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

## Acknowledgements

The authors would like to thank all patients and families who contributed to this project. This project was funded through the following grants awarded to the respective investigators, for which we are grateful: Clinician Scientist Awards to DWTL (NMRC/CSA-INV/0025/2017, MOH-000375-01), DSWT (NMRC/CSA/MOH-CSAINV19NOV-0005) and NGI (NMRC/CSA/001/2016, MOH-000325-00), the Peter Fu Head and Neck Cancer Program (under the Oncology Academic Clinical Program, National Cancer Centre Singapore) to NGI. Both nivolumab and ipilimumab were supported generously by Bristol-Myers Squibb (USA) and ONO Pharmaceuticals (Taiwan); both did not participate in study design, data collection, and analysis, or manuscript writing.

## Author contributions

D.W.T.L. designed clinical investigation, interpreted the data, and wrote the manuscript. N.G.I. oversaw the project, interpreted data, and wrote the manuscript. H.F.K. designed clinical protocol and provided samples. L.S., C.H.L., S.H.T., and H.S.Q. performed experiments, analyzed, and interpreted the data. D.S.W.T., E.H.T., W.L.T., A.J., B.C.G., M.L.K.C., B.C.L., Q.S.N., R.L.H., M.K.A. provided feedback in clinical investigation and data interpretation. J.N.L. and F.W.Y.T. performed immuno-fluorescence validation. J.P.S.Y. provide expertize and resources for immunofluorescence analysis. All authors reviewed and contributed to the manuscript.

## Competing interests

NGI sits on the Scientific Advisory Boards of PairX Therapeutics, Ver-Immune and Vivo Surgical, and has received honoraria/funding from Merck, Kalbe Biotech and Agilent, all of which are outside the scope of this submitted work. D.W.T.L. has received research funding from Bristol-Myers Squibb through institution. He has also received honoraria/travel support from Merck, Roche, Boehringer-Ingelheim, Taiho Pharmaceuticals, Pfizer, Novartis, and Eisai which are outside the scope of the submitted work. The remaining authors declare no other competing interests.
