## [Peer Review File · Nature Communications]

Clinical efficacy and biomarker analysis of dual PD-1/ CTLA-4 blockade in recurrent/metastatic EBV-associated nasopharyngeal carcinomaREVIEWER COMMENTS

Reviewer #1 (Remarks to the Author): with expertise in head and neck Cancer, clinical

Comments for authors:

Line 34. The statement in the abstract that low EBV DNA pts “ showed better response and progression-free survival.” Is not supported by statistically robust data provided, and the authors recognize as much on response on line 140 and beyond by stating it only “ trended” the abstract needs to reflect the reality of lack of power on this .

Line 59. Radiation alone is not the standard of care for locally advanced NPC. Please rephrase to avoid this misreading.

Line 60 Now GC plus ICI is the standard for first line relapsed / metastatic disease. This present standard has significant implications for the application of information from this study and needs to be addressed.

Line 133. Why was baseline DNA available only for the first 26 patients? This could influence any interpretations of this marker because of an unsuspected bias of some sort. As a prospective secondary endpoint this should have been collected on all patients. The study design suggests that this was even an eligibility criterion. Similarly, why was mIHC done on only 17 samples of pretreatment bx if there were 22 total samples? Such numbers discrepancies throughout need to be addressed.

Line 148: It is unclear what Fig 2 b is supposed to represent, given the statement in the text;” For these analyses, response was based on specific response at the site where the tissue was obtained, rather than best overall response (Fig 2b)” Are we to assume that fig 2 b only represents response at one tumor site as the text says? If so the overall labelling of fig 2 is misleading. Fig 2 should represent overall classical RECIST response. Not just biopsied tumor response. If this is incorrect, the text should be reworded.

Line 177: $8 + 5 = 13$ but above it says there were 19 on treatment samples. What about the remaining 6? Were these the SD?

Lines 211 and elsewhere . A recurrent theme is that assays on baseline tumor tissue did not correlate with outcome, but assays performed on “ on treatment “ samples did. This seems to be blended in with T cell subpopulation assays at baseline. This distinction could be clarified. The authors, however for this and the EBV DNA baseline observation, do not grapple with the challenge of what amounts to “ more likely versus less likely” data and whether the magnitude of enrichment for nonresponders is enough to justify withholding treatments, for example.

Line 253; the data for EBV DNA levels distinguishing responders from non responders is not a very strong correlation and should not be construed to suggest that the lower rr seen for high ebv dna is actionable, that is, reason not to give ICI.

It is surprising that in the figure one data and elsewhere there is no mention of just what the prior systemic therapies were nor how many were patients relapsing after chemo radiation or after first line systemic treatment for r/m disease.

Reviewer #2 (Remarks to the Author): with expertise in head and neck cancer, clinical, cancer immunotherapy

This manuscript describes a prospective phase 2 trial of 40 patients with nasopharyngeal cancer treated with Nivo plus ipi. The manuscript nicely combines clinical endpoints of response with extensive correlative data including pre-treatment and on treatment biopsy specimens and analyses . However a number of difficulties and areas of substantial lack of clarity must be addressed to enable appropriate interpretation of this report. First they indicate that there was no correlation of response with expression of PD - L1. It is unclear to what extent the trial is powered for this association, based on other data in nasopharyngeal cancer or other head and neck tumours. Furthermore it is unclear

whether they used TPS or CPS scoring. This is crucial to put their data in perspective since this is standard of care to test for extent of PD-L1 positivity. Next they indicate that deep immunophenotyping of tumor biopsies showed activation of an adaptive immune response, but it is unclear whether this was correlated with evidence of pathologic response or radiographic response. This is key for any biomarker. Next the authors repeatedly indicate that this was a single arm non-randomized phase 2 study. It is obvious that a single arm study is non-randomized, and so they should eliminate this redundancy in the text. Next, little to no information is given as to how the dosing of the two therapies was arrived at or if there was a dose finding phase. Much controversy has arisen in the past year or two regarding combination of nivolumab and ipilimumab in the clinic, and what is the appropriate dose and schedule of ipilimumab in particular. Further justification for selection of this regimen is necessary, particularly given the negative Checkmate 651 trial, using the same regimen as is reported here which was reported a year ago at ESMO. Next the author should comment on the lack of complete responses, since other trials even with nivolumab or pembrolizumab alone have shown at least some complete responses. Further information on the role of EBV-DNA would be important as to whether this is simply a prognostic marker of viral load or truly predicts benefit of the immunotherapy. Was there a correlation with duration or depth of response? Next although it is highly appealing that paired pre-treatment and on-treatment biopsies exist the authors do not maximize their utility comparing static baseline biomarkers versus dynamic ones which change during treatment. More should be made of this and the figures do not clearly segregate when the on-treatment biopsies are being analyzed with biomarkers. Next figure 2 panel B seems to show three hyper-progressing patients. Biomarkers should be segregated for whether we can identify those patients before treatment. Next this trial is focused on patients with prior exposure to chemotherapy, and that should be more clearly defined, in particular whether there was an association of immunotherapy response based on prior gemcitabine or cisplatin or cetuximab as some other trials in H&N cancer have shown for PD-1 responsiveness. In that regard the manuscript Introduction references the Keynote 048 trial, but this was in the first line R/M setting and this is not correct here, since the prior chemotherapy-treated patients in this trial would more accurately relate to the Checkmate-141 and Keynote-040 trials. The references should be updated to reflect the role of immunotherapy in large phase 3 trials of chemotherapy resistant head and neck cancer.

Reviewer #3 (Remarks to the Author): with expertise in biostatistics, clinical trial study design

Statistical analysis was well covered in the manuscript, including trial design based on Simon two-stage design, descriptive analysis of patient characteristics, efficacy, and toxicity, association analysis of EBV-DNA with response, and evaluation of mIHC/IF with response association. The use of lesion level (site) response is appropriate, instead of overall response.

Below are the statistical comments for clarification to strengthen the manuscript.

1. Please clarify what type of Simon two-stage design was used. While the main text indicated optimal design, the protocol used minimax design.
2. What is the distribution of PD1+/CTLA4- cell counts and the distribution of CD8+ cell counts for the PD1+/CTLA4- cells? Similarly, what is the distribution of PD1-/CTLA4+ cell counts and the distribution of CD8+ cell counts for the PD1-/CTLA4+ cells? Are the cells abundant? If total number of cells is small, would the analysis be less robust (e.g., denominator is 10 PD1+/CTLA4- cells with 5 of them with CD8+, leading 50% CD8+)?
3. Will analysis of on-treatment samples cause biased results without adjusting for pre-treatment data?
4. Would ECOG and prior radiotherapy be associated with clinical outcomes (e.g., response, PFS, and OS)? If yes, would these factors be adjusted in biomarker analysis?
5. Biomarker analysis was based on specific response at the site where the tissue was obtained, rather than best overall response. What was the distribution of response in lesion level (site) and how was the response defined in lesion level?
6. What is association of EBV-DNA with PD1+/CTLA4-/CD8+ and PD1-/CTLA4+/CD8+?

Response to Reviewers

We thank the reviewers for a thorough review of our manuscript and suggestions for improvement. We have found the reviewers' comments to be very helpful and constructive. The manuscript has been revised addressing reviewers' concerns.

Reviewer #1

Reviewer Comment 1.1	Line 34. The statement in the abstract that low EBV DNA pts “ showed better response and progression-free survival.” Is not supported by statistically robust data provided, and the authors recognize as much on response on line 140 and beyond by stating it only “ trended” the abstract needs to reflect the reality of lack of power on this .
Author Response 1.1	We agree with the reviewer and have revised the abstract in line with the reviewer's comments.
Changes to Manuscript	Ln 32-34: There was no correlation of response with PD-L1 expression or tumor mutation burden, however patients with low plasma circulating EBV-DNA titre (<7,800 IU/ml) showed a trend to better response and progression-free survival.

Reviewer Comment 1.2	Reviewer comments: Line 59. Radiation alone is not the standard of care for locally advanced NPC. Please rephrase to avoid this misreading.
Author Response 1.2	We thank the reviewer for the opportunity to make a clarification. We have rephrased the line to leave no doubt.
Changes to Manuscript	Ln 59: It is highly responsive to radiotherapy and chemotherapy ³ , and concurrent chemotherapy and radiotherapy is the standard of care for locally advanced disease ⁴⁻⁶ .

Reviewer Comment 1.3	Line 60 Now GC plus ICI is the standard for first line relapsed / metastatic disease. This present standard has significant implications for the application of information from this study and needs to be addressed.
Author Response 1.3	We thank the reviewer for the insightful comments and thorough review of our manuscript. We have made revised the introduction to address the reviewer's comments, as well as acknowledged the limitation of the interpretation of this data in the discussion.
Changes to Manuscript	Ln 60-64: Up till 2022 gemcitabine and cisplatin (GC) was standard first line treatment for recurrent/metastatic (R/M) NPC ⁷ . However, this is likely to change pending regulatory approvals of PD-1 inhibitors from China. Pivotal phase III studies utilizing a GC backbone and a partner PD-1 inhibitor have demonstrated superior progression-free survival (PFS) benefit over GC and placebo, although overall survival (OS) remains immature ^{8,9} Line 350-353: Finally, given the introduction of PD-1 inhibitors into the 1 st line treatment space, we do not know the activity of this combination in patients who had prior exposure to checkpoint inhibitor.

Reviewer Comment 1.4	Line 133. Why was baseline DNA available only for the first 26 patients? This could influence any interpretations of this marker because of an unsuspected bias of some sort. As a prospective secondary endpoint this should have been collected on all patients. The study design
---

	suggests that this was even an eligibility criterion. Similarly, why was mIHC done on only 17 samples of pre-treatment bx if there were 22 total samples? Such numbers discrepancies throughout need to be addressed.
Author Response 1.4	We thank the reviewers for their comments. We have detailed which samples were used for which test in a new Supplementary Table 4. Baseline plasma EBV DNA was available for all patients. However, we have reported only the initial 26 patient dataset here, precisely because we wanted to study this early observed phenomenon further. The subsequent 14 patients from this expansion are included in a further expansion up to 80 patients to fully understand the impact of the observed difference in the first 26 patients, hence the results were not reported here. mIHC was only done on 17 samples instead of 22 total samples because of limitations of sample quantity and quality. All data available on mIHC has been presented.
Changes to Manuscript	No changes to manuscript but added supplementary Table 4.

Reviewer Comment 1.5	Line 148: It is unclear what Fig 2b is supposed to represent, given the statement in the text;” For these analyses, response was based on specific response at the site where the tissue was obtained, rather than best overall response (Fig 2b)” Are we to assume that fig 2 b only represents response at one tumor site as the text says? If so the overall labelling of fig 2 is misleading. Fig 2 should represent overall classical RECIST response. Not just biopsied tumor response. If this is incorrect, the text should be reworded.
Author Response 1.5	The response reflected in Fig 2b is classical RECIST response. We have changed the figure legend to reflect that. However, the reviewer is correct to understand that the exploratory biomarker analyses reported are based on tumor response of the biopsied site, as indicated in the text.
Changes to Manuscript	Changes to Figure 2b legend.
Reviewer Comment 1.6	Line 177:8+ 5 =13 but above it says there were 19 on treatment samples. What about the remaining 6? Were these the SD?
Author Response 1.6	We thank the reviewer for the comment. The remaining 6 is SD. We only compared responders and progressors as defined by site-specific radiological response in the biopsied lesion.
Changes to Manuscript	NA
Reviewer Comment 1.8	Lines 211 and elsewhere . A recurrent theme is that assays on baseline tumor tissue did not correlate with outcome, but assays performed on “ on treatment “ samples did. This seems to be blended in with T cell subpopulation assays at baseline. This distinction could be clarified. The authors, however for this and the EBV DNA baseline observation, do not grapple with the challenge of what amounts to “ more likely versus less

	likely” data and whether the magnitude of enrichment for nonresponders is enough to justify withholding treatments, for example.
Author Response 1.8	We agree with the reviewer that the magnitude of the differences we see based on baseline plasma EBV DNA and other tumor assays are not enough to effect a change in practice. We have further edited the language in the manuscript to reflect the above. We have generated a plot to visually illustrate the magnitude of changes seen in the pre-treatment vs on-treatment biopsies (supplementary figure 4b).
Changes to Manuscript	New supplementary figure 4b
Reviewer Comment 1.9	Line 253; the data for EBV DNA levels distinguishing responders from non responders is not a very strong correlation and should not be construed to suggest that the lower rr seen for high ebv dna is actionable, that is, reason not to give ICI.
Author Response 1.9	We agree with the reviewers comments. We have expounded this in our discussion and have further emphasized in the text that this observation needs further validation.
Changes to Manuscript	
Reviewer Comment 1.10	It is surprising that in the figure one data and elsewhere there is no mention of just what the prior systemic therapies were nor how many were patients relapsing after chemo radiation or after first line systemic treatment for r/m disease.
Author Response 1.10	We thank the reviewer for their comment, and we have amended Figure 1 and provided further details in Supplementary Table 1 to reflect this.
Changes to Manuscript	Addition of prior systemic therapies, relapse after chemoradiation, and after 1 st line therapy appended in Supp Table 1

Reviewer #2

Reviewer Comment 2.1	First they indicate that there was no correlation of response with expression of PD - L1. It is unclear to what extent the trial is powered for this association, based on other data in nasopharyngeal cancer or other head and neck tumours. Furthermore it is unclear whether they used TPS or CPS scoring. This is crucial to put their data inter perspective since this is standard of care to test for extent of PD-L1 positivity.
Author Response 2.1	We thank the reviewer for the comment. We assessed PD-L1 expression from Nanostring mRNA assay, and this showed no correlation to response. The trial is not powered to show this association by this or IHC testing. PD-L1 testing is not a standard of care in NPC in Asia with no standardized assay, and multiple studies have also demonstrated no association with CPI treatment. We have elaborated and clarified this aspect in the text and discussion.
Changes to Manuscript	Ln 303-307: Conventional PD-L1 expression does not correlate well with response to PD-1/PD-L1 blockade in NPC ^{14,17} , unlike in non-small cell lung or head and neck squamous cell cancers ^{18,19} . As this study was not powered to show this association, we did not proceed to do PD-L1 IHC given the limited tissue available

	from the biopsies which were being subjected to other biomarker analyses. PD-L1 mRNA expression on Nanostring was done but it did not correspond to treatment outcomes.
Reviewer Comment 2.2	Next they indicate that deep immunophenotyping of tumor biopsies showed activation of an adaptive immune response, but it is unclear whether this was correlated with evidence of pathologic response or radiographic response. This is key for any biomarker
Author Response 2.2	We agree with the reviewers comments. The deep immunophenotyping signal was associated with site-specific radiographic response in the biopsied lesion as mentioned in line 190.
Changes to Manuscript	No changes
Reviewer Comment 2.3	Next the authors repeatedly indicate that this was a single arm non-randomized phase 2 study. It is obvious that a single arm study is non-randomized, and so they should eliminate this redundancy in the text.
Author Response 2.3	We thank the reviewer for this astute observation. We have eliminated the redundant text.
Changes to Manuscript	Redundant text removed
Reviewer Comment 2.4	Next, little to no information is given as to how the dosing of the two therapies was arrived at or if there was a dose finding phase. Much controversy has arisen in the past year or two regarding come binding nivo and ipi in the clinic, and what is the appropriate dose and schedule of ipi in particular. Further justification for selection of this regimen is necessary, particularly given the negative Checkmate 651 trial, using the same regimen as is reported here which was reported a year ago at ESMO.
Author Response 2.4	We thank the reviewer for comments. The dose used here was based on Scott Antonio's initial phase I study published in JCO in 2018. This is appended and elaborated in the text. To further contextualize the outcomes seen here we have added the need for further studies comparing this dual IO combination with chemotherapy in the discussion

Changes to Manuscript	(line 379-381): The schedule and doses of this combination were adopted from initial phase I studies done in solid tumors and confirmed in expanded cohorts in lung cancer ^{32,33} Ln 285-290: To set this in context, in reported studies of similar CTLA4/PD-1 inhibitor combinations in head and neck squamous cancer, this was not superior to standard EXTREME combination chemotherapy ^{36,37} , suggesting possible limitations of a checkpoint inhibitor only strategy. In a relatively more chemotherapy-sensitive tumor like NPC, a formal comparison of chemotherapy and dual checkpoint inhibitors compared to chemotherapy and PD-1 inhibitors in R/M NPC would be required.
Reviewer Comment 2.5	Next the author should comment on the lack of complete responses, since other trials even with nivo or Pembro alone have shown at least some complete responses.
Author Response 2.5	The lack of complete responses seen here is possibly due to the small sample size, or a difference in tumor biology of this EBV-related tumor in response to dual immunotherapy as opposed to monotherapy. We have included this comment in the text.
Changes to Manuscript	In 274-276: In contrast to reported single agent studies, where one or two complete responses were reported, we did not see any in this study. This is likely a function of sample size, although a difference in the biological response to dual immunotherapy in NPC cannot be excluded
Reviewer Comment 2.6	Further information on the role of EBVDNA would be important as to whether this is simply a prognostic marker of viral load or truly predicts benefit of the immunotherapy. Was there a correlation with duration or depth of response?
Author Response 2.6	We thank the reviewer for the comment. We have further analysed the time on treatment, and depth of response and there remains a trend to correlation of better clinical outcomes with the EBV viral load. This is presented in the supplementary figure and further elaborated in the text. We also acknowledge in Line 186-187 the limitations and constraints of this observed association, and is the motivation for a further expansion cohort to study this observation.
Changes to Manuscript	Ln 148-150: Correlation to the depth of response and time on therapy also trended better in the EBV _{low} cohort (Fig. 3b and Supplementary Table 4)
Reviewer Comment 2.7	Next although it is highly appealing that paired pre-treatment and on treatment biopsies exist the authors do not maximize their utility comparing Static baseline biomarkers versus dynamic ones which change during treatment. More should be made of this and the figures do not clearly segregate when the on treatment biopsies are being analyzed with biomarkers.
Author	We agree that an in-depth analysis of biomarker changes in pre- vs on-treatment samples would reveal important insights into response dynamics. Unfortunately,

Response 2.7	the sample size of our current study limits the scope and complexity of analyses we can perform- ideally, we would fit a unified model with interaction terms and investigate interactions between gene expression and timepoint. To investigate dynamics in this study, we constructed biomarkers derived from pre-treatment and on-treatment samples and compared the genes and their associations in each dataset (Supplementary Figure 4). We demonstrate the performance of the pre-treatment signature in on-treatment samples and the on-treatment signature in pre-treatment samples. We looked at associations between changes in gene expression and treatment response to elucidate dynamics associated with response. We added an analysis of cell-type changes derived from nanostring data (Figure 5a and Supplementary Figure 5). These results demonstrate that changes in the immune microenvironment are associated with response and only become evident after the start of treatment. Future work in larger cohorts is needed to further investigate these dynamics and reveal their utility in biomarker construction. We have also revised the figure legends and references to signatures to clarify which biopsies were used to derive each.
Changes to Manuscript	Changes to figure legends and reference to Supplementary Figure 4
Reviewer Comment 2.8	Next figure 2 panel B seems to show three hyper progressing patients. Biomarkers should be segregated for whether we can identify those patients before treatment.
Author Response 2.8	We thank the reviewer for the comment. Out of 3 hyper progressive patients, there is only 1 patient who had available tumor for analysis, and we did not find any clear difference in the gene expression analysis when compared to normal progressors.
Changes to Manuscript	Line 272-274: Of note, 3 patients appeared to hyperprogress (Fig. 2B), however only one patient had sufficient tissue for biomarker analysis, and we were unable to elucidate further possible mechanisms for this phenomenon.
Reviewer Comment 2.9	Next this trial is focused on patients with prior exposure to chemotherapy, and that should be more clearly defined, in particular whether there was an association of Immunotherapy response based on prior gemcitabine or cisplatin or cetuximab as some other trials in H&N cancer have shown for PD-1 responsiveness.
Author Response 2.9	We thank the reviewer for the comment. We have delineated the prior chemotherapy regimens and the association with response to this combination and there is no association seen. We have appended the data in supplementary table 3.

Changes to Manuscript	Supplementary Table 3
Reviewer Comment 2.10	In that regard the manuscript Introduction references the Keynote 048 trial, but this was in the first line R/M setting and this is not correct here, since the prior chemotherapy-treated patients in this trial would more accurately relate to the Checkmate-141 and Keynote-040 trials. The references should be updated to reflect the role of Immunotherapy in large phase 3 trials of chemotherapy resistant head and neck cancer.
Author Response 2.10	We thank the reviewer for the comment. We have reflected this differences in the introduction as suggested in line 69-70.
Changes to Manuscript	Line 69-70: However, unlike in R/M head and neck squamous cancers ^{16,17} , this modest efficacy in NPC has not resulted in a survival benefit compared to chemotherapy ^{18,19} .

Reviewer #3

Reviewer Comment 3.1	Please clarify what type of Simon two-stage design was used. While the main text indicated optimal design, the protocol used minimax design.
Author Response 3.1	We thank the reviewer for the comment. The reviewer's astute observation is correct, we have amended the text to reflect minimax design.
Changes to Manuscript	"optimal" to "minimax" (line 508)
Reviewer Comment 3.2	What is the distribution of PD1+/CTLA4- cell counts and the distribution of CD8+ cell counts for the PD1+/CTLA4- cells? Similarly, what is the distribution of PD1-/CTLA4+ cell counts and the distribution of CD8+ cell counts for the PD1-/CTLA4+ cells? Are the cells abundant? If total number of cells is small, would the analysis be less robust (e.g., denominator is 10 PD1+/CTLA4- cells with 5 of them with CD8+, leading 50% CD8+)?
Author Response	We thank the reviewer for the comment. The table is appended below showing that the cell counts are abundant and robust.

nse 3.2	Patient ID	Response	CD8	PD.1	CTLA4	PD.1+CTLA4 ⁺ CD8 ⁺	PD.1-CTLA4 ⁺ CD8 ⁺	PD1 ⁺ CTLA4 ⁺ CD8 ⁺ /CD8 (%)	PD1 ⁻ CTLA4 ⁺ CD8 ⁺ /CD8 (%)
	003	PD	945	35	3264	14	298	1.48	31.53
	007	PD	2511	1968	11179	205	255	8.16	10.16
	019	PD	6447	32	3530	8	1103	0.12	17.11
	020	PD	2801	117	8227	8	1467	0.29	52.37
	001	PR	2985	1832	1626	878	207	29.41	6.93
	004	PR	2070	408	730	218	47	10.53	2.27
	010	PR	895	508	1855	95	120	10.61	13.41
	012	PR	7215	4450	4903	1511	311	20.94	4.31
	013	PR	1556	1848	2917	469	71	30.14	4.56
	014	PR	2348	782	1572	336	116	14.31	4.94
	029	PR	1103	205	154	47	0	4.26	0.00
	002	SD	5404	529	1524	377	55	6.98	1.02
	008	SD	4565	503	1454	190	166	4.16	3.64
	018	SD	4492	2135	1017	893	56	19.88	1.25
	021	SD	2150	43	6	34	2	1.58	0.09
	032	SD	958	72	195	46	51	4.80	5.32
033	SD	743	119	1300	17	134	2.29	18.03	
Changes to Manuscript									
Reviewer Comment 3.3	Will analysis of on-treatment samples cause biased results without adjusting for pre-treatment data?								
Author Response 3.3	We thank the reviewer for the comment. The model we fit for differential expression includes all samples, both pre- and on-treatment, and accounts for timepoint effects. We extract the estimates for the comparison of interest as necessary. We additionally investigated pre-treatment estimates and identified few significant associations (Supplementary Figure 4). To compare pre- and on-treatment associations with response in more detail, we examined on-treatment signatures in pre-treatment samples and pre-treatment signatures in on-treatment samples. We also investigated associations between response and changes in gene expression between the time points. Together, the results from these analyses suggest that predictive associations are strongest in on-treatment samples even after accounting for pre-treatment effects.								
Changes to Manuscript	Changes to figure legends and reference to Supplementary Figure 4								

Reviewer Comment 3.4	Would ECOG and prior radiotherapy be associated with clinical outcomes (e.g., response, PFS, and OS)? If yes, would these factors be adjusted in biomarker analysis?
Author Response 3.4	We thank the reviewer for the comment. The clinical factors mentioned were not associated with clinical outcomes (Supplementary table 3)
Changes to Manuscript	Added supplementary table 3.
Reviewer Comment 3.5	Biomarker analysis was based on specific response at the site where the tissue was obtained, rather than best overall response. What was the distribution of response in lesion level (site) and how was the response defined in lesion level?
Author Response 3.5	We thank the reviewer for the comment. The majority of the biopsies were taken from the primary nasopharyngeal site or lymph node metastases and detailed in supplementary table 4 shown with associated response.
Changes to Manuscript	Added supplementary table 4
Reviewer Comment 3.6	What is association of EBV-DNA with PD1+/CTLA4-/CD8+ and PD1-/CTLA4+/CD8+?
Author Response 3.6	We thank the reviewer for the comment. There is no correlation between EBV-DNA and PD1+CTLA4-CD8+ or PD1-CTLA4+CD8+ population, and we have supplemented this in Supplementary Fig 5b.

	  Correlation EBV PD1+CTLA4- $r = 0.006135$ $p\text{-value} = 0.9830$   Correlation EBV PD1-CTLA4+ $r = 0.1362$ $p\text{-value} = 0.5997$  
Changes to Manuscript	Added supplementary figure 5b

REVIEWERS' COMMENTS

Reviewer #1 (Remarks to the Author):

none

Reviewer #2 (Remarks to the Author):

The authors have satisfactorily address my comments and concerns

Reviewer #3 (Remarks to the Author):

Statistical concerns have been addressed, except the one below.

It is claimed that “two subpopulations (PD-1+CTLA-4-CD8+, PD-1-CTLA-4+CD8+) appeared to be independent of each other (Supplementary Figure 5b)”. While the Supplementary Figure 5b is not available, intuitively, when one subpopulation grows, the other subpopulation will decrease, as shown in the rebuttal letter of response to reviewer 3 comment 3.2 with a table showing negative correlation of both subpopulations, especially for PR vs PD. Please clarify it.

Response to Reviewers

We thank the reviewers for a thorough review of our manuscript and suggestions for improvement. We have found the reviewers' comments to be very helpful and constructive. The manuscript has been revised addressing reviewers' concerns.

Reviewer #1: None

Reviewer #2: None

Reviewer #3

Reviewer Comment 3.1	Statistical concerns have been addressed, except the one below. It is claimed that “two subpopulations (PD-1+CTLA-4-CD8+, PD-1-CTLA-4+CD8+) appeared to be independent of each other (Supplementary Figure 5b)”. While the Supplementary Figure 5b is not available, intuitively, when one subpopulation grows, the other subpopulation will decrease, as shown in the rebuttal letter of response to reviewer 3 comment 3.2 with a table showing negative correlation of both subpopulations, especially for PR vs PD. Please clarify it.
Author Response 3.1	The populations are independent of each other as shown in the correlation graph below. PD-1-CTLA4- population contributes to the remaining CD8 population.   Correlation PD1+CTLA4- vs PD1-CTLA4+ CD8 Population   Correlation PD1+CTLA4- vs PD1-CTLA4+ CD8 Population  
Changes to Manuscript	Figure added to Supplementary Figure 5b, Supplementary Figure 5b moved to 5c.